# Prognostic Markers and Driver Genes and Options for Targeted Therapy in Human-Papillomavirus-Positive Tonsillar and Base-of-Tongue Squamous Cell Carcinoma

**DOI:** 10.3390/v13050910

**Published:** 2021-05-14

**Authors:** Anders Näsman, Stefan Holzhauser, Ourania N. Kostopoulou, Mark Zupancic, Andreas Ährlund-Richter, Juan Du, Tina Dalianis

**Affiliations:** 1Department of Oncology-Pathology, Karolinska Institutet, Bioclinicum J6:20, Karolinska University Hospital, 171 64 Stockholm, Sweden; Anders.Nasman@ki.se (A.N.); Stefan.Holzhauser@ki.se (S.H.); Ourania.Kostopoulou@ki.se (O.N.K.); Mark.Zupancic@ki.se (M.Z.); Andreas.Ahrlund@gmail.com (A.Ä.-R.); 2Department of Microbiology, Tumor Biology and Cellular Biology, Karolinska Institutet, Biomedicum, 171 77 Stockholm, Sweden; Juan.Du@ki.se

**Keywords:** human papillomavirus, tonsillar squamous cell carcinoma, base-of-tongue squamous cell carcinoma, oropharyngeal squamous cell carcinoma, mutations, prognostic marker, driver genes, targeted therapy

## Abstract

The incidence of Human-papillomavirus-positive (HPV^+^) tonsillar and base-of-tongue squamous cell carcinoma (TSCC and BOTSCC, respectively) is increasing epidemically, but they have better prognosis than equivalent HPV-negative (HPV^−^) cancers, with roughly 80% vs. 50% 3-year disease-free survival, respectively. The majority of HPV^+^ TSCC and BOTSCC patients therefore most likely do not require the intensified chemoradiotherapy given today to head and neck cancer patients and would with de-escalated therapy avoid several severe side effects. Moreover, for those with poor prognosis, survival has not improved, so better-tailored alternatives are urgently needed. In line with refined personalized medicine, recent studies have focused on identifying predictive markers and driver cancer genes useful for better stratifying patient treatment as well as for targeted therapy. This review presents some of these endeavors and briefly describes some recent experimental progress and some clinical trials with targeted therapy.

## 1. Introduction

In 2007, the International Agency for Research against Cancer (IARC) recognized human papillomavirus (HPV) as a risk factor, besides smoking and alcohol consumption, for oropharyngeal squamous cell carcinoma (OPSCC) and especially its specific subsite tonsillar squamous cell carcinoma (TSCC) [1,2,3,4]. Since then, a significant rise in the incidences of HPV-positive (HPV^+^) TSCC and base-of-tongue squamous cell carcinoma (BOTSCC), the two main HPV^+^ OPSCC subsites, has been seen in many countries [5,6,7,8,9,10,11,12,13,14,15,16,17,18]. Notably, patients with HPV^+^ TSCC or BOTSCC have better prognosis than those with equivalent HPV-negative (HPV^−^) OPSCC and head and neck squamous cell carcinoma (HNSCC), with roughly 80% vs. 50% disease-free survival, respectively [1,2,3,4,8,16]. Treatment of HNSCC, including OPSCC, has, however, in the past years become more aggressive with induction chemo- or concurrent chemoradiotherapy (and sometimes targeted EGFR inhibitor (cetuximab) instead of chemotherapy) [19,20,21]. This intensified therapy, with more side effects, is likely an overtreatment for most patients with HPV^+^ TSCC and BOTSCC and has unfortunately not improved the outcome for those with poor prognosis. Therefore, better-tailored therapy is definitely required [1,2,3,4,8,16,19,20,21,22,23,24,25]. To enable the stratification of patients for de-escalated or targeted therapy, recent research has focused on exploring the role of HPV^+^ status in survival per OPSCC subsite and identifying predictive biomarkers and driver genes optimal for targeted therapy [26,27,28,29,30,31,32,33,34,35,36,37,38,39,40,41,42,43,44,45,46,47,48,49,50]. Below, efforts to reach these goals as well as an introduction to some experimental progress and some clinical trials with targeted therapy for these specific cancers are presented.

## 2. Special Features of HPV^+^ Compared to HPV^−^ TSCC, BOTSCC, and Other OPSCC Subsites, the Epidemic Increase, and the Definition of HPV^+^ Status

### 2.1. The Presence of HPV Is Most Evident in TSCC and BOTSCC, the Two Major OPSCC Subsites, and There Is No Doubt about Its Prognostic Value in These Two Subsites

An association between HPV, OPSCC, and TSCC was reported in 2000, and with radiotherapy and surgery, these HPV^+^ cases were shown to have much better prognosis than equivalent HPV^−^ cases (80% vs. 40–50% 5-year survival, respectively) [3,4]. Shortly after, in 2004, a similar association of HPV with BOTSCC, but not to mobile tongue cancer, was disclosed, with analogous better prognosis for HPV^+^ as compared to HPV^−^ BOTSCC [2]. It has since then been verified in a plethora of reports that HPV prevalence is much higher in TSCC and BOTSCC, the two major lymphoepithelial- and crypt-containing subsites of OPSCC, as compared to that in other OPSCC subsites, e.g., the soft palate and the pharyngeal wall [51,52,53,54]. More specifically, in a systematic review of 58 distinct cohorts, the prevalence of HPV was 56% in TSCC/BOTSCC as compared to 19% at other OPSCC subsites [51]. Furthermore, the favorable prognostic value of HPV in TSCC and BOTSCC has been confirmed repeatedly and by now, there is no doubt regarding it, while the prognostic value of HPV at other OPSCC subsites accounting for 10–20% of all OPSCC has not really been verified so far [1,2,3,4,52,53,54,55,56]. Notably, in the *American Joint Committee on Cancer (AJCC) 8th Edition*, the distinction of HPV^+^ OPSCC into different subsites was not considered, and this is, in our opinion, of some concern, especially when conducting and evaluating clinical trials [57,58]. Furthermore, according to the AJCC 8th Edition, an HPV^+^ status is defined by p16 ^INK4a^ (p16) overexpression, which also adds to the uncertainty (see Section 2.4 below).

### 2.2. The Epidemic Increase in the Incidence of HPV^+^ OPSCC Is Most Evident in HPV^+^ TSCC and BOTSCC, and There Are Changes in Patient Age over Time

In the past decades, there has been a sharp rise in the incidence of OPSCC in many countries [5,6,7,8,9,10,11,13,14,15,16,18,59,60,61,62,63]. It is also of note that this surge is primarily due to an increase in HPV^+^ cases, mainly HPV^+^ TSCC and BOTSCC [5,6,7,8,9,10,11,13,14,15,16,18,59,60,63]. This is exceptionally clear in countries were the statistics are presented separately for different OPSCC subsites [12,14,59,60,61,63]. Importantly, the epidemic rise in the incidence of HPV^+^ TSCC and BOTSCC emphasizes the urgency of optimizing therapy for this growing group of patients [61]. Furthermore, despite the efforts that are made to vaccinate against HPV, it will take decades before there will be a decline in its incidence [61]. Notably, however, the mean age of the patients has changed over time [2,3,7,10,12,14,15,62]. Decades earlier, patients with HPV^+^ cancer were on average 5–10 years younger than those with HPV^−^ cancer. However, recently, this discrepancy has almost disappeared, which is also of significance when striving to better individualize therapy [2,3,7,8,10,12,14,15,62].

### 2.3. Some General Differences between HPV^+^ and HPV^−^ TSCC and BOTSCC

Today, HPV^+^ and HPV^−^ TSCC and BOTSCC are acknowledged as separate entities, with different characteristics, where the former is linked to the oncogenic capacity of high-risk (HR) HPVs and the latter is mainly associated with smoking and alcohol consumption [1,2,3,4]. HPV^+^ TSCC and BOTSCC generally have normal p53, overexpress p16, are often aneuploid, are less differentiated, present lymph node metastasis early on, and often have cervical-cancer-like amplification of chromosome 3q, while HPV^−^ TSCC and BOTSCC mostly lack these characteristics [3,28,40,49,64,65,66,67]. In addition, phosphatidylinositol-4,5-bisphosphate 3-kinase catalytic subunit alpha (*PIK3CA)* and fibroblast growth factor 3 (*FGFR3*) mutations, both targetable, often occur in HPV^+^ but not in HPV^−^ cancer, where p53 mutations dominate [28,40,49,67]. Still, irrespective of differentiation, or ploidy, etc., HPV^+^ TSCC and BOTSCC patients have better prognosis than those with corresponding HPV^−^ cancers [1,2,3,4,64,65,66,67].

### 2.4. Definition of HPV^+^ Status in OPSCC, TSCC, and BOTSCC

HPVs are circular double-stranded DNA viruses with a genome of almost 8000 kb, arbitrarily divided into a regulatory, an early, and a late region and enclosed in a 52–55 nm virion [67]. Regulatory proteins E1–E2, E4–E7 are encoded for by the early region, and virus capsid proteins L1 and L2 are encoded for by the late region [68]. In HR-HPVs, E6 and E7 are regarded as oncogenes, in which E6 binds to p53 and causes its degradation, in this way preventing control of DNA damage, cell repair, and apoptosis, while E7 binds to Rb and abrogates and deregulates cell cycle control [68]. The latter also triggers overexpression of the cyclin-dependent kinase inhibitor p16^INK4a^ (p16) [68].

The golden standard for an active oncogenic HPV infection is therefore the expression of HPV E6 and E7 mRNA, but since this was in the past more difficult to assay for on a regular basis, overexpression of p16 in >70% of the tumor cells was instead used as a surrogate marker for the presence of HPV in OPSCC [68,69]. The concordance between HPV E6 and E7 mRNA expression and p16 overexpression is, however, not complete, since around 10–15% of TSCCs and BOTSCCs expressing p16 are not HPV^+^, and this discrepancy is even greater at other OPSCC subsites and in other HNSCCs [51,52,53,54,69]. However, it has been shown especially in TSCC and BOTSCC, as well as in OPSCC, that the presence of HPV DNA together with the overexpression of p16 is almost equivalent to the presence of the golden standard [69,70].

## 3. Treatment of Tonsillar and Base-of-Tongue Cancer

### 3.1. Treatment of HPV^+^ and HPV^−^ Tonsillar and Base-of-Tongue Cancer and TNM-8

As already mentioned above (in Section 2.1), according to the *AJCC 8th Edition*, a distinction between p16-overexpressing OPSCC and no p16 overexpression is done, not considering the distinction of OPSCC subsites. Optimally, for better accuracy, based on data described above, one would limit this distinction to TSCC and BOTSCC and use HPV E6 and E7 mRNA expression or the presence of p16 overexpression together with the presence of HPV DNA to indicate HPV^+^ status (in Section 2.4). This would allow for a more stringent follow-up, especially when conducting clinical trials. However, this is not always applied today, and for this reason, some of the clinical data on HPV^+^ OPSCC should be taken with caution.

### 3.2. Treatment of HPV^+^ and HPV^−^ TSCC and BOTSCC

Similar to HNSCC and OPSCC in general, most patients with TSCC and BOTSCC present symptoms when their tumors are relatively large, and especially in HPV^+^ TSCC and BOTSCC, lymph node metastasis upon diagnosis is common [19,20,21]. Initially, treatment consisted of radiotherapy and subsequent surgery, if necessary, or surgery and subsequent radiotherapy [2,3,4,20]. Today, depending also on the tumor burden, patients are frequently given radiotherapy with doses up to 70 Gray (Gy), along with induction or concomitant chemotherapy, and sometimes also epidermal growth factor receptor (EGFR) inhibitors [19,20,21,22,71,72]. This treatment may in some cases, and more frequently in the past, also be accompanied by modified neck node surgery in patients with lymph node metastasis remaining after radiotherapy. Furthermore, in HNSCC patients with advanced/unresectable tumors, the possibility to add immunotherapy with PD-1 inhibitors upfront, alone or together with chemotherapy, has been attempted and has shown positive effects on survival [73]. Upon recurrence, salvage surgery or re-irradiation, with or without chemotherapy, is sometimes possible, and nowadays, immunotherapy with PD-1 inhibitors is attempted [72]. The administration of palliative treatment is always, as for all other treatments, dependent on the patient’s performing status, and chemotherapy and/or PD-1 inhibitors are often used in parallel with painkillers [72].

Acute side effects come along with the intensive treatment above, such as difficulty in swallowing and eating, caused by radiation, in addition to nausea and mucositis, accompanied by considerable weight loss, as well as local and systemic infections and fatigue [19,20,21,22,71,72,74,75,76]. Moreover, when chemoradiotherapy is followed by modified neck node surgery due to lymph node metastasis remaining after treatment, it may lead to additional side effects [74,75,76]. This together with previous radiotherapy may lead to increased fibrosis and stiffness of the neck, which can worsen the swallowing difficulties, and patients may also present reduced mobility of the shoulder.

Long-term side effects also occur, and many patients suffer from xerostomia, alterations in taste, and continued difficulties with swallowing. Some patients encounter trismus and a hearing deficit, which may worsen [71,72,74,75,76]. In addition, depression is a side effect that should be followed up and not neglected [77,78]. Radio-osteonecrosis is also a late side effect and needs reconstruction surgery [71,72,74,75,76].

### 3.3. De-Escalation of Treatment of HPV^+^ OPSCC

As frequently mentioned above, HPV^+^ TSCC, BOTSCC, and OPSCC have been shown to have much better prognosis than corresponding HPV^−^ cancer, and due to the plethora of side effects, also indicated above, efforts to de-escalate therapy have been made. For example, attempts have been initiated to reduce radio- or chemotherapy [72,79,80]. In ECOG 1308, a study on stage III–IV OPSCC using p16 overexpression and HPV in situ hybridization for HPV^+^ status, early results suggested that a response to induction chemotherapy is correlated with a low 1-year failure rate upon reduced-dose radiation [72]. In non-inferiority RTOG 1016, a multi-center US trial, and a similar phase III De-ESCALaTE HPV clinical trial, randomizing patients on cetuximab or cisplatin with concurrent accelerated intensified radiation therapy showed that the former is inferior to the latter [72,79]. Moreover, in a different report, it was shown that de-escalation of radiotherapy is associated with worse prognoses in patients with *PIK3CA* mutations than those harboring wild-type *PIK3CA* [26].

The latter study discloses the importance of accumulating more knowledge on the biological significance of different biomarkers upon conducting de-escalated therapy or embarking on different types of targeted therapies. Several studies have recently made attempts to use different clinical characteristics and biomarkers to estimate which patients would be eligible for de-escalated therapy [23,29,47]. However, as the latter study shows, the situation is complex and more information is needed [23,29,47]. Below, attempts to find prognostic markers will be presented.

## 4. The Search of Prognostic or Targetable Biomarkers in HPV^+^ and HPV^−^ TSCC, BOTSCC, and OPSCC by Immunohistochemistry or Viral Gene Expression

The first attempts to find biomarkers in this category of tumors were in order to characterize whether there were specific differences between HPV^+^ and HPV^−^ OPSCC, as described above (in Section 2.3). These studies revealed some similarities between HPV^+^ TSCC and BOTSCC and HPV^+^ cervical and vulvar cancer, such as the frequent absence of a p53 mutation and/or the amplification of parts of chromosome 3q, but neither had a prognostic impact [64,81,82]. The specific search for prognostic or targetable markers was embarked on somewhat later. Initial reports explored the nature of the viral genome and markers possible to study by immunohistochemistry (IHC). Subsequently, molecular methods, e.g., sequencing and proteomic analyses, became much more commonly used, and some of these different categories of undertakings will be described below.

### 4.1. Early Studies on the Presence of HPV DNA, p16, p53, and Smoking and Survival

Many studies focused on HPV DNA, p16, p53, and smoking in OPSCC and survival, but here, only a few will be mentioned [23,24,70,83]. Early on, it was shown that non-smokers with HPV-DNA-positive TSCC or OPSCC or p16-overexpressing OPSCC had better prognosis than corresponding patients who smoked [24,83]. However, this was likely due to the fact that having either HPV-DNA-positive or p16-overexpressing OPSCC was a suboptimal way of defining an HPV^+^ tumor, since although the two correlated in most cases, there still was a 10–15% discrepancy between the two [69]. More recently, it has been shown that when requiring the presence of HPV DNA and p16 overexpression to define HPV-positive status, the significance of being a smoker or a non-smoker is not as obvious [23,70]. An extensive number of studies have also focused on the role of p53 in HPV^+^ TSCC, BOTSCC, and/or OPSCC, and it has by now repeatedly been shown that p53 mutations are less frequent in HPV^+^ TSCC, BOTSCC, and/or OPSCC as compared to their HPV^−^ counterparts [28,49,84]. Moreover, importantly, the presence of p53 mutations in HPV-DNA-positive or p16-overexpressing or HPV-DNA-negative and p16-overexpressing TSCC/BOTSCC or OPSCC has so far not been shown to be associated with worse prognosis [28,49,84].

### 4.2. Studies on the Physical Status of the HPV Genome; Viral Load; Viral Gene Expression in TSCC, BOTSCC, and OPSCC; and Correlation with Disease-Free Survival

The physical status of HPV in TSCC, BOTSCC, and OPSCC has been examined in different ways and reported to be episomal or integrated or both [85,86,87,88]. However, a specific physical state of the virus could not be correlated with survival, while a high viral load was shown or tended to correlate with a better clinical outcome [85,86,87,88]. Furthermore, the presence of HPV16 E2 mRNA, which could be expected to be correlated with the presence of an episomal HPV genome, although this does not have to be the case, was also correlated with better prognosis, as also reported in cervical cancer [88,89]. In this context, it should be mentioned that in addition to HPV16 E2 mRNA, E5 and E7 mRNA expression and correlation with survival were also examined due to the potential influence of E5 and E7 on MHC class I expression and because E2 and E5 are sometimes lost upon viral genome integration [89]. In that report, it was demonstrated that HPV16 E2, E5, and E7 mRNA are not correlated with MHC class I expression [89].

### 4.3. Studies on Immune Cells in HPV^+^ and HPV^−^ TSCC, BOTSCC, and OPSCC and Survival

Immunological and stem cell markers have been examined extensively, and are still being examined, by IHC and by other means in HPV^+^ and HPV^−^ OPSCC, TSCC, and BOTSCC [33,34,35,36,37,38,39,40,41,42,45,46,48,50]. Moreover, recently, an extensive review on the OPSCC tumor microenvironment with regard to options for immunotherapy presented a detailed overview of immune cells present in the microenvironment and their potential role in the promotion of active or suppressive immune responses in OPSCC [90]. Most reports agree that the numbers of CD8^+^ lymphocytes infiltrating or surrounding the tumor are generally higher in HPV^+^ TSCC, BOTSCC, and OPSCC than in respective HPV^−^ tumors and that having high numbers of CD8^+^ cells is correlated with a better outcome irrespective of the HPV status [23,38,39,40,41,47,90]. The prognostic value of CD4^+^ and FoxP3^+^ lymphocytes has also been examined, but in two studies, neither correlated alone with the outcome. However, independent of the HPV status, having a low CD4^+^/CD8^+^ ratio or a high CD8^+^/FoxP3^+^ ratio is associated with improved survival [38,39]. In another study, however, it was shown that a high frequency of Tbet+ Tregs is associated with prolonged disease-specific survival and this was most likely because their presence showed a high numbers of effector T-cells [90,91]. The role of macrophages was also discussed extensively in the same review [90]. CD68^+^ CD163^+^ M2 macrophages were often observed in OPSCC, and a high infiltration of these macrophages was correlated with poor prognosis in two studies on HNSCC, with most or all cases being OPSCC [92,93,94]. In this context, the role of checkpoint inhibitors was also examined in OPSCC. In one study, PD-L1 expression as a biomarker was examined in CD68^+^ macrophages, and in HPV^−^ OPSCC, increased numbers of CD68^+^ macrophages expressing PD-L1 suggested a favorable immune milieu [40]. In another study, likewise, in HPV^+^ tumors infiltrated by CD8^+^ and CD68^+^ immune cells with high PD-L1 expression, overall survival was favorable [95].

### 4.4. Studies on the Role of Major Histocompatibility Complex (MHC) Antigen in HPV^+^ and HPV^−^ TSCC, BOTSCC, and OPSCC and Survival

The expression of antigens of the major histocompatibility complex (MHC) in TSCC and BOTSCC has also been examined, especially since it is known that HPV16 E5 and E7 can downregulate MHC antigens [96,97,98]. Notably, in some studies on HPV^+^ TSCC and BOTSCC, absent/low MHC class I expression was of a positive prognostic value, while the reverse may have been anticipated, which was also observed in patients with HPV^−^ cancer [35,36]. In contrast to MHC class I, MHC class II expression was not shown to have an impact on the clinical outcome in TSCC and BOTSCC irrespective of the HPV status [35]. Following up on these findings, TAP1 and TAP2 and LMP2, LMP7, and LMP10 expression, all influencing MHC peptide antigen processing, was also examined [45,46]. It was then disclosed that TAP2, LMP2, and LMP7 expression often declined in HPV^+^ cancer and that decreased LMP7 expression and medium/high nuclear LMP10 staining correlated with better survival in HPV^+^ cancer [45,46]. Hypothetically, it should be expected that downregulation of MHC class I, and components of the antigen processing machinery, is of evolutionary benefit for HPV infection as well as for HPV^+^ cancer in terms of evading the effects of the immune system. However, HPV^+^ tumors have also been suggested to be more sensitive to radiotherapy than HPV^−^ tumors, and that could be part of the explanation for why patients with HPV^+^ TSCC/BOTSCC/OPSCC fare better [99,100,101]. It is unclear, however, whether this really has been studied extensively. Nevertheless, in vitro, it has been experimentally shown that upon irradiation, in some cases, there is an increase in MHC class I expression, while in contrast, HPV16 E5 mRNA is decreased, thereby potentially increasing the immune sensitivity of the tumors [102]. Achieving increased immune sensitivity of HPV^+^ tumors would be favorable for the clinical outcome and in concordance with experimental data obtained in vivo [103]. In the latter study, HPV^+^ tumors were curable after cisplatin or radiation therapy only in mice with a normal immune system and not in immune-incompetent mice, and it was therefore suggested that to combat these tumors, an immune response was required [103].

### 4.5. Studies of Various Cell Markers, Such as Stem Cell and Suppressor Gene Markers, Defined by IHC in HPV^+^ and HPV^−^ TSCC, BOTSCC, and OPSCC and Survival

A plethora of other markers, of which some have stem cell or tumor suppressor character, have also been studied by IHC for their possible predictive role in survival in HPV^+^ TSCC, BOTSCC, and OPSCC, and here, only a few are described [33,34,42,104]. Early on, it was shown that having low CD44 intensity expression or CD98 expression was correlated with better survival irrespective of the HPV status in the tumors, while for tumor suppressor genes *LRIG1-3*, only high LRIG1 expression was correlated with a better clinical outcome [33,34,42]. More recently, having low psoriasin expression was correlated with better survival in HPV^+^ BOTSCC [50].

To summarize, immunological and general IHC analysis shows that some markers are specific for only HPV^+^ or HPV^−^ TSCC, BOTSCC, or OPSCC while many are common for both HPV^+^ and HPV^−^ tumors, as well as for many other tumors.

## 5. The Search for Prognostic or Targetable Biomarkers in HPV^+^ and HPV^−^ TSCC, BOTSCC, and OPSCC by Molecular Methods and the Potential Use of Targeted Therapy

In recent years, molecular technology has improved considerably, allowing for the possibility to perform mutational analysis by both next-generation sequencing (NGS) of exomes and whole-genome sequencing, and the costs of these analyses have also decreased. Furthermore, importantly, it is possible to do so with low amounts of DNA and even with lower-quality DNA from formalin-fixed-paraffin-embedded (FFPE) material. RNA sequencing and protein profiling have also been performed, but here and definitely for the latter, fresh frozen material is of benefit.

### 5.1. DNA Sequencing, Mutation Analysis, and Possibilities for Targeted Therapy in HPV^+^ and HPV^−^ TSCC, BOTSCC, and OPSCC

Using NGS in HNSCC, several different features have been disclosed between HPV^+^ and HPV^−^ cancers, including the potential of targeting some of them [28,43,49,67,105,106,107,108,109]. HPV^+^ TSCC, BOTSCC, or OPSCC mainly presented mutations in the *PIK3CA*, notch homolog 1 translocation-associated *(NOTCH1)*, and *FGFR3* genes, while HPV^−^ tumors mainly had mutations in *TP53* and cyclin-dependent kinase inhibitor 2A/B (*CDKN2A/B)* [28,43,49,67,104,105,106,107,108,109,110]. The prognostic values of some of these genes have been reported, but there are discrepancies between the studies, and the data should be taken with caution. For example, in one study, *TP53* was shown to be of prognostic value in HPV^−^ OPSCC, but this could not be confirmed in two other studies [28,49,110]. Mutated *FGFR3,* especially the *FGFR3* p.S249C variant, was also correlated with worse prognosis, similar to that in cervical cancer in one study, but this could not be confirmed in HPV^+^ TSCC and BOTSCC in a subsequent report [28,67,111,112]. Mutated *PIK3CA* has in this context not generally been shown to have a prognostic value, but in one report, it was shown to have a favorable prognostic value, while in another study, with de-intensified radiotherapy, it was correlated with worse survival [26,28,43,46,49]. Irrespective of whether the mutated genes are prognostic, the fact that some mutations are present could potentially increase the sensitivity of these tumors to targeted therapy. Two mutant p53-re-activating compounds APR-246 and COTI-2 are currently in use in clinical trials, but their efficacy needs to be further evaluated [113,114]. There are, however, several inhibitors targeting *PIK3CA* and *FGFR3*, and recently, the Food and Drug Administration (FDA) approved the phosphoinositide 3-kinase inhibitor (PI3K inhibitor) alpelisib (BYL719) for clinical use for advanced breast cancer and the FGFR inhibitor erdafitinib (JNJ-42756493) for advanced bladder cancer [115,116,117]. The latter two types of inhibitors may well be of special importance for HPV^+^ TSCC, BOTSCC, and OPSCC, since often more than 20% and often 10%, respectively, of such tumors harbor *PIK3CA* and/or *FGFR3* mutations [28,49].

To conclude, gene sequencing has disclosed important information about relevant mutations, where targeted therapy could be an important option.

### 5.2. MicroRNA Expression in HPV^+^ and HPV^−^ TSCC, BOTSCC, and OPSCC in Relation to the Clinical Outcome

There are many reports on microRNA (miR) expression in HPV^+^ and HPV^−^ TSCC, BOTSCC, and/or OPSCC, of which some are mentioned here [30,31,118,119,120,121,122,123,124]. The data are variable, with few miRs found common between studies, and not all analysis have taken the HPV status into account. However, miR-9, miR-155, and miR-163b were in some reports shown to be overexpressed in HPV^+^ as compared to HPV^−^ OPSCC, while miR-31 and miR-193b were downregulated [31,118,119,120,122]. Recently, eight meta-signature miRs were identified as potential biomarkers of oropharyngeal cancers, but in that study, the correlation of these miRs with the HPV status was not reported [118]. Instead, in a separate study, not necessarily related to TSCC, BOTCC, or OPSCC, the expression of miR was analyzed as a function of HPV E6 and E7 mRNA expression in human foreskin keratinocytes [125]. There is clearly a large variation between the different publications, and reasons for the variations between the studies could be the assessment of large miR numbers in a limited number of samples and that validation cohorts were lacking. A small number of studies have examined miR expression in TSCC, BOTSCC, or OPSCC in association with HPV and survival, and here also, the specific miRs differ [30,31,118]. In one study, overexpression of miR-142-3p, miR-146a, and miR-26b was associated with better overall survival while the expression of miR-31, miR-24, and miR-193b was negatively correlated with outcome [118]. In another report, miR-146a, miR-155, and miR-200b expression was correlated with specific treatments and clinical outcome in HNSCC, which included a cohort of OPSCC and miR-146a and miR-155 were suggested to serve as surrogate markers for immune cell infiltration [30]. We have also found a positive correlation with survival in patients with HPV^+^ TSCC and BOTSCC with high miR-155 expression, while the opposite was observed for those with high miR-185 expression, and in that report, miR-193b expression was not correlated with outcome [27].

To summarize, the number of publications on miR expression in relation to the clinical outcome and HPV status in TSCC, BOTSCC, and OPSCC is still limited, and the large variation in the data reported so far suggests that additional information is necessary before miRs can be exploited clinically.

### 5.3. Analysis of the Transcriptome in HPV^+^ and HPV^−^ TSCC, BOTSCC, and OPSCC

Studies on the transcriptome of HPV^+^ and HPV^−^ TSCC, BOTSCC, and OPSCC can be of various kinds. There are reports describing the specific expression of HPV mRNAs in HNSCC, while others focus on all types of mRNA. Some of these former studies have been mentioned above [89,97,98]. Others have disclosed differences between HPV^+^ and HPV^−^ TSCC, BOTSCC, and OPSCC with regard to immune responses, apoptosis, proliferation and the cell cycle, and the immune defense [123,126,127]. The latter would be expected since the nature of the two types of tumors as well as their microenvironments are so different, as also described above (Section 4.3 and Section 4.4).

### 5.4. Protein Profiling in HPV^+^ and HPV^−^ TSCC, BOTSCC, and OPSCC

There are still limited numbers of reports on protein profiling in TSCC, BOTSCC, and OPSCC, of which three are mentioned here [43,44,128]. One report used reverse-phase protein array profiling on a restricted set of 137 total and phosphorylated proteins and disclosed differences between HPV^+^ and HPV^−^ cancers in, e.g., the PI3K/protein kinase B (AKT)/mammalian target of rapamycin (mTOR) receptor kinase pathways [43]. Protein expression was also evaluated in relation to *PIK3CA* mutations in HPV^+^ OPSCC and was found to be associated with the activation of mTOR, and the authors suggested that mTOR inhibitors could be of use in *PIK3CA*-mutated HPV^+^ OPSCC [43]. Another study used proteomic profiling by mass spectrometry and disclosed differences between HPV^+^ and HPV^−^ OPSCC in many pathways, of which one was increased ASS1 expression in HPV^+^ OPSCC [35]. More recently, we examined the expression of 167 immune-related proteins using two Olink multiplex immunoassays in material derived from fresh frozen samples from 42 HPV^+^ and 17 HPV^−^ TSCC and BOTSCC in relation to normal tissue [128]. In that report, some proteins tended to be associated with survival, and for HPV^+^ TSCC and BOTSCC, these proteins were mainly related to angiogenesis and hypoxia [128]. Notably, a high expression of vascular endothelial growth factor A (VEGFA) was correlated with poorer prognosis in HPV^+^ cancer, and we suggested that angiogenesis-related proteins may serve as potential targets for therapy in HPV^+^ TSCC and BOTSCC [128].

### 5.5. Microbiome Studies in HPV^+^ and HPV^−^ TSCC, BOTSCC, and OPSCC

The oral microbiome is emerging as a promising biomarker of head and neck cancers [129]. An early study found that HNSCC patients have a significant lower diversity of microbiota species than healthy controls and different bacterial taxonomies can be used for distinguishing oral cancer samples from OPSCC and normal samples [130]. Moreover, bacteria species, including *Fusobacterium nucleatum* and *Actinobacteria*, which are often found to contribute to carcinogenesis in in vitro experiments, are often observed in greater abundance in the HNSCC oral microbiome [129,130,131]. Notably, although there are large variations in the significant changed bacteria, similar functional signatures, such as enrichment of pro-inflammatory features, are demonstrated to be enriched in oral squamous cell carcinoma patients [129].

It has also been noted that HPV infection in the cervix is associated with microbiota changes, as has also been reported in HPV-related cervical intraepithelial neoplasia (CIN) and cervical cancer [132,133]. However, the studies on microbiota and HPV infection in HNSCC are limited. There are some bacteria found to be more abundant in HPV^+^ HNSCC, but more large well-designed cohorts on the microbiome and HNSCC are urgently needed, especially considering and including HPV status [130]. It is, namely, possible that HPV-infection-associated shifts in the composition of the microbial community may contribute to the progression of HPV^+^ TSCC, BOTSCC, and OPSCC. Whether manipulating oral microbiota can be used for improving clinical outcome is of great potential interest.

## 6. Targeted Therapy in HPV^+^ and HPV^−^ TSCC, BOTSCC, and OPSCC; Some Experimental Progress; and Clinical Trials

So far, apart from the use of immunotherapy, i.e., PD-1 inhibitors, we are not aware that targeted therapy has been used extensively or compassionately specifically in HPV^+^ and HPV^−^ TSCC, BOTSCC, or OPSCC [73]. However, as mentioned above, both PI3K and FGFR inhibitors have been FDA approved for use in advanced breast cancer and urinary bladder cancer, respectively [116,117]. Nonetheless, there are clinical trials ongoing, targeting the PI3K/AKT/mTOR pathway in HNSCC, with the hope that, e.g., PI3K inhibitors would be more efficient in cancers where the PIK3CA gene is mutated [134]. In addition, others and we have also attempted to experimentally explore the efficacy of such targeted therapy, alone or in different combinations, on different types of solid tumor cell lines, of which some were derived from HNSCC or more specifically from TSCC or BOTSCC. In Section 6.2, some of these attempts are briefly presented. However, before going into these attempts in more detail, it should be mentioned that other approaches are also ongoing for targeted therapy in HNSCC, of which some target the regulation of NFkB signaling by ubiquitination [135]. Here, drugs such as proteosome inhibitors, e.g., bortezomib or inhibitors of apoptosis proteins (IAP), such as Debio-113, are discussed [135]. However, the exact importance and the specific effects of these drugs in HPV^+^ OPSCC need to be explored further.

### 6.1. Clinical Trials with PI3K, mTOR, and FGFR Inhibitors in HNSCC and Other Solid Tumors

PIK3CA encodes p110alpha protein, which is a catalytic subunit of the class I PI 3-kinases (PI3K) and is a frequently altered oncogene in human cancers, including HNSCC [115,134]. A number of clinical studies targeting PI3K in HNSCC have been initiated, but none of these have specifically targeted HPV^+^ TSCC, BOTSCC, and OPSCC [115,134]. Buparlisib (BKM120) has been used in a monotherapy clinical trial (NCT01737450) in patients with metastatic head and neck cancer, recurrent or progressive (PIK-ORL). Others (NCT02145312, NTC01602315, NCT02051751, and NCT03138070) have used the FDA-approved alpelisib (BYL719) in recurrent or metastatic HNSCC. Everolimus (RAD001), an FDA-approved mTOR inhibitor, is presently included in clinical trials (NCT01111058, NCT01051791, and NCT01283334) on recurrent or metastatic HNSCC.

To the best of our knowledge, there are no specific FGFR inhibitor studies on TSCC, BOTSCC, or OPSCC. There is, however, a phase 1 clinical study NTC01703481 evaluating the safety, pharmacokinetics, and pharmacodynamics of the FDA-approved FGFR inhibitor erdafitinib (JNJ-42756495) in patients with solid tumors, in which patients with HNSCC are included.

Many of the above, but not all, of the clinical trials are still ongoing, but it is impossible to distinguish any positive effects specifically for HPV^+^ TSCC, BOTSCC, and OPSCC. Nevertheless, positive effects with prolonged progression-free survival with some of the above treatments have been documented for other types of solid tumors, such as for breast cancer, but there are also considerable side effects accompanying these treatments [116,117,136,137,138,139,140].

For example, in breast cancer patients treated with the PI3K inhibitor alpelisib (BYL719) and fulvestrant (ICI-182780, ZD 9238, ZM 182780), hyperglycemia, rashes, and diarrhea were frequently encountered; in some cases, pneumonitis and interstitial lung disease and, in other cases, deaths have been reported [116,137]. In many cases, the side effects could be treated, but this was not always the situation and targeted therapy had to be abandoned, and as with all drugs, the benefits must outweigh the side effects.

Patients with urinary bladder cancer treated with FGFR inhibitors frequently experienced hyperphosphatemia, constipation, decreased appetite, stomatitis, dry mouth, elevated creatinine, asthenia, and fatigue [138]. Although benefits from treatment could be observed, in many cases, treatment had to be terminated, here also due to toxicity, and moreover, resistance was frequently observed upon mono-treatments [138].

In metastatic breast cancer, to avoid rapid resistance development, PI3K inhibitors, anti HER2, PI3K, and CDK4/6 inhibitors are being used clinically in combination and also explored experimentally [136,141]. Consequently, to avoid some of these side effects, i.e., to be able to decrease the inhibitor doses and to avoid resistance development, some of these inhibitors have been combined. One clinical trial, e.g., combined the CDK4/6 inhibitor palbociclib (PD-0332991) with the PI3K/mTOR inhibitor gedatolisib (PF-05212384) for patients with advanced squamous cell lung, pancreatic, head and neck, and other solid tumors. Below, some experimental targeted therapies, including combinations of some of the above inhibitors, for HNSCC, as well as TSCC and BOTSCC, will be described.

### 6.2. Experimental Targeted Therapy in HPV^+^ and HPV^−^ TSCC, BOTSCC, and OPSCC

There have been a few experimental studies performed specifically for HPV^+^ and HPV^−^ TSCC, BOTSCC, OPSCC, and HNSCC with targeted therapy. Below, some are described. In addition, studies including other types of solid tumors are briefly mentioned for comparison.

In 2015, the effects of the mTOR inhibitor everolimus (RAD001), the protein kinase inhibitor sorafenib (BAY 43-9006), and the multi-targeted tyrosine kinase inhibitor sunitinib (SU11248) were examined in two p16^−^ HNSCC cell lines and in an HPV^+^ cervical cancer line, all shown to express mTOR and amphiregulin (AREG) [142]. Notably, however, the HPV^+^ cervical cancer cell line had higher mTOR and lower AREG expression as compared to the p16-negative cell lines [142]. The applied drugs decreased mTOR expression, which could be of benefit for delaying tumor progression. In addition, everolimus (RAD001) decreased AREG independent of HPV status, while sorafenib and sunitinib increased AREG expression in the HNSCC cell lines but not in the HPV^+^ cervical cancer cell line [142]. The authors concluded that further studies would be necessary to evaluate HPV-dependent therapy responses and the potential consequences for options for therapy.

In a contemporary report on the latter, six HNSCC cell lines, of which two had *PIK3CA* mutations and four did not, were exposed to the PI3K inhibitor alpelisib (BYL719) [143]. It was then disclosed that the two cell lines with *PIK3CA* mutations were generally, but not completely, more sensitive to alpelisib (BYL719) and had lower IC_50_ values than the PIK3CA wild-type cell lines [143]. To the best of our knowledge, however, none of these cell lines were HPV^+^.

Prior to the above study, a pre-clinical evaluation of the dual mTOR-PI3K inhibitor dactolisib (BEZ235) was performed in nasopharyngeal cancer models, showing that the inhibitor induced dose-dependent responses in all cell lines and that having a *PIK3CA* mutation was not needed for a response [144]. However, there was no obvious synergistic effect when dactolisib (BEZ235) was combined with chemotherapy [144].

More recently, we tested the effects of the PI3K inhibitor dactolisib (BEZ235) and the FGFR inhibitor AZD4547 on the HPV^+^ TSCC cell line UPCI-SCC-154 and the HPV^−^ UT-SCC-60A cell line and found dose-dependent responses despite the fact that none of the cell lines had a *PIK3CA* mutation [145]. Furthermore, when the two inhibitors were combined, enhanced responses were disclosed [145]. The data suggest that the concentrations of the two inhibitors could be decreased and that resistance development to the individual drugs likely would decrease when they are used together [145].

The findings in our report are analogous to the findings in ovarian cancer cell lines, where mTOR inhibitors (rapamycin, sirolimus) and FGFR inhibitors (infigratinib, BGJ398) were combined and, upon combining, the inhibitors enhanced inhibitory effects on cell growth [146]. In addition, inhibitory effects were observed on migration, and cell cycle arrest and apoptosis were also observed [146]. Experiments were also performed in vivo, and while treating the mice with one drug alone did not stop tumor growth, inhibition of tumor growth was readily observed upon combination of the inhibitors [146].

More recently, we continued our efforts on HPV^+^ and HPV^−^ TSCC and BOTSCC cell lines, with and without *PIK3CA* and *FGFR3* mutations, and used the FDA-approved PI3K and FGFR inhibitors alpelisib (BYL719) and erdafitinib (JNJ-42756493), respectively [147]. In that report, we found that the cells responded in a dose-dependent way to single-inhibitor treatments, and synergy was disclosed upon combining the two inhibitors, independent of whether the cell lines had *PIK3CA* or *FGFR* mutations [147]. Furthermore, when combining the inhibitors with cisplatin or docetaxel, neutral, additive, or rarely antagonistic effects were disclosed [147]. The latter was similar to what was disclosed in [147].

CDK4/6 inhibitors have, to the best of our knowledge, not been tested extensively experimentally in HNSCC. However, in one report, the CDK4/6 inhibitor abemaciclib (LY2835219) was combined with metformin and enhanced activity was observed in HNSCC cell lines [148]. Likewise, when combining the CDK4/6 inhibitor abemaciclib (LY2835219) with an mTOR inhibitor, synergy was found both in vitro and in vivo in HNSCC cell lines [149]. In another report, however, it was noted that cisplatin exposure caused c-Myc-dependent resistance to CDK4/6 inhibition in HPV^−^ HNSCC cell lines [150].

To summarize, many in vitro and in vivo experimental studies on targeted therapy are conducted in many solid tumors today, and with time, extensive knowledge will also accumulate on HPV^+^ TSCC and BOTSCC. Hopefully, these studies and other clinical trials will bring the field forward and offer better-tailored therapy, with increased survival and quality of life for patients with these diseases in the future.

## 7. Concluding Remarks

HPV^+^ TSCC, BOTSCC, and OPSCC have only been acknowledged as specific entities since 2007. However, despite this short time of perception, much knowledge has accumulated on their biological nature and many biomarkers have been disclosed, some of which will be useful for prognostic prediction and others for directing targeted therapy. Moreover, today, there are vaccines that can prevent HPV16 infection, the most common HPV type in the above tumors. Hopefully, taken together, the accumulated knowledge in this field will improve therapy for this growing patient group, whether it is de-escalated or targeted therapy, and will result in increased patient survival and quality of life, as well as in the long run possible extinction of this debilitating disease.

## Data Availability

Not applicable.

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
