# Peer review of "Prognostic Markers and Driver Genes and Options for Targeted Therapy in Human-Papillomavirus-Positive Tonsillar and Base-of-Tongue Squamous Cell Carcinoma"

_viruses, 2021, doi:10.3390/v13050910_

Round 1

Reviewer 1 Report

The authors stress the common practice that treatment of HNSCC, including OPSCC, is more aggressive with more side effects and irrespective of the tumor type (OPSCC vs other anatomical sites; HPV+ vs HPV-). A more personalized strategy is warranted. Therefore, prognostic and driver genes can be very effective to be used for targeted therapy. The review not only summarizes some of the genes but also the immune infiltrate as prognostic markers. Therefore it would be better to rephrase the title to better describe the content of the review.

Other major comments:

  1. In section 2.3. some general differences are given between HPV+ and HPV- TSCC and BOTSCC, however, as the review title includes prognostic and driver genes the reader would expect also an overview of differential expression of genes between HPV+ and HPV- Orophayngeal patients, which might be interesting for targeting therapy. There are publications (among which Reder et al., Oral Oncology 2019; Smeets et al., Cellular Oncology 2009; Yao et al., JITC 2020), including from the cancer genome atlas network (Nature 2015) about the gene expression in head and neck cancer and even focused on oropharyngeal cancer.
  2. In section 4.3. the correlation between high numbers of intratumoral CD8+ T cells and clinical outcome is mentioned, while the prognostic value of CD4+ is not shown. In oropharyngeal cancer it has been demonstrated that the high numbers of CD4+ T cells present in the tumor is correlating with better survival and that this was often accompanied by Foxp3+ Tbet+ Tregs (see the review indicated by reference 90 and publications herein).
  3. Section 4.4. lowest part (lines 288-294) would better fit in section 4.2. to prevent repeat.
  4. The authors mentioned the pitfalls of the various studies but it would be nice when the authors give a design of the study to be able to get the answers on the questions and testing the hypothesis.

Minor comments:

  1. Abstract page 1, line 22: and as well as…please remove “as well as”.
  2. Introduction page 1, line 30: oropharyngeal- and tonsillar squamous cell carcinoma….to my knowledge is tonsillar squamous cell carcinoma one of the tumors belonging to oropharyngeal cancer.
  3. Introduction: It would be helpful to shortly describe the search criteria for finding the literature reviewed in this manuscript.
  4. Typo in section 5.1 line 340: as should be replaced by was. Also in section 5.2 line 376 the word “to” is missing between “correlated outcome [117]”.
  5. Section 6.1 lines 486 – 508 concerns a large part of the side effects of the PI3K and EGFR inhibitors. Would it be possible to write this in a more concise manner ?

Author Response

Dear Reviewer,

We thank you for your valuable comments and have revised the manuscript accordingly point by point. 

- The authors stress the common practice that treatment of HNSCC, including OPSCC, is more aggressive with more side effects and irrespective of the tumor type (OPSCC vs other anatomical sites; HPV+ vs HPV-). A more personalized strategy is warranted. Therefore, prognostic and driver genes can be very effective to be used for targeted therapy. The review not only summarizes some of the genes but also the immune infiltrate as prognostic markers. Therefore it would be better to rephrase the title to better describe the content of the review.

-We thank the reviewer for this comment and have now changed the title to:

Prognostic markers and driver genes and options for targeted therapy in tonsillar and base of tongue squamous cell carcinoma.

This way the immune infiltrates and other stem cell markers are included under the term “markers”.

Other major comments:

  1. In section 2.3. some general differences are given between HPV+ and HPV- TSCC and BOTSCC, however, as the review title includes prognostic and driver genes the reader would expect also an overview of differential expression of genes between HPV+ and HPV- Orophayngeal patients, which might be interesting for targeting therapy. There are publications (among which Reder et al., Oral Oncology 2019; Smeets et al., Cellular Oncology 2009; Yao et al., JITC 2020), including from the cancer genome atlas network (Nature 2015) about the gene expression in head and neck cancer and even focused on oropharyngeal cancer.

Section 2.3. is meant to be a very short introduction, to that HPV+ OPSCC (or more specifically HPV+ TSCC and HPV+ BOTSCC) are different entities from HPV- OPSCC, and here only some few differences between the two are pinpointed in this section. We just wanted to emphasize the fact that the entities are different. Today, the scientific community widely, accepts that these entities differ, but this was not the case some 15 years ago, and it was only in 2007, that the International Agency of Cancer Research acknowledged this (reference 3 in the manuscript). In addition, this review focuses on HPV+ OPSCC and more specifically HPV+ TSCC and BOTSCC, that are increasing in incidence, and where new treatment options are urgently needed.

The references the reviewer refers to are interesting papers, but are therefore slightly off the actual focus of this specific paragraph.

-Reder et al. Oral Oncology 2019, describes differences in genes in HPV+ OPSCC with and without recurrences. (i.e. not relevant for section 2.3).

-Smeets et al. Cellular Oncology 2009, has the aim to to investigate patterns of chromosomal aberrations of only HPV-negative oral and oropharyngeal squamous cell  carcinomas (OOSCC) (altogether 39 tumors) in order to improve stratification of patients regarding outcome. (i.e. not relevant for section 2.3)

Yao et al. 2020 JITC, focuses on immune related genes in head and neck cancer and is indeed an interesting paper. Together the TCGA cohort (Nature 2015) and the TMA cohort includes 30 oropharyngeal cancer cases, but these are relatively limited. However, although the authors do present HPV+ head and neck cancer and data on immune related genes, they do not specify, what is the case specifically for oropharyngeal cancer. (i.e. not detailed enough for section 2.3.)

  1. In section 4.3. the correlation between high numbers of intratumoral CD8+ T cells and clinical outcome is mentioned, while the prognostic value of CD4+ is not shown. In oropharyngeal cancer it has been demonstrated that the high numbers of CD4+ T cells present in the tumor is correlating with better survival and that this was often accompanied by Foxp3+ Tbet+ Tregs (see the review indicated by reference 90 and publications herein).

The text has now been modified accordingly and an additional reference has been referred to.

3.Section 4.4. lowest part (lines 288-294) would better fit in section 4.2. to prevent repeat.

We thank the reviewer for this comment and the text has been modified accordingly.

4.The authors mentioned the pitfalls of the various studies but it would be nice when the authors give a design of the study to be able to get the answers on the questions and testing the hypothesis.

This is a very general comment, and we are not sure as to what specifically the reviewer is referring to. Scientific work shows progress and limitations. We have tried to present the field today with its progress and some limitations. It is was not our aim to scrutinize each study to suggest an improved designs of each subfield to move the general field further. Although this would be an interesting pursuit per se, it would be beyond the scope of this review. Nonetheless, we are confident that colleagues in the different subfields will improve their specific niches.

Minor comments:

  1. Abstract page 1, line 22: and as well as…please remove “as well as”.

We thank the reviewer for this comment and as suggested this has been done.

  1. Introduction page 1, line 30: oropharyngeal- and tonsillar squamous cell carcinoma….to my knowledge is tonsillar squamous cell carcinoma one of the tumors belonging to oropharyngeal cancer.

We thank the reviewer for this comment and the sentence has now been re-phrased.

  1. Introduction: It would be helpful to shortly describe the search criteria for finding the literature reviewed in this manuscript.

There were no general search criteria for finding the literature at the initiation of writing this manuscript since it is not a systematic review, and therefore this has not been described in the introduction. However, within each paragraph, many of the headings, as well as the specific genes were used as search criteria. In addition, upon reading the literature within each search, additional interesting references were also scrutinized and some were referred to.

  1. Typo in section 5.1 line 340: as should be replaced by was. Also in section 5.2 line 376 the word “to” is missing between “correlated outcome [117]”.

We thank the reviewer for these comments and have corrected the text.

  1. Section 6.1 lines 486 – 508 concerns a large part of the side effects of the PI3K and EGFR inhibitors. Would it be possible to write this in a more concise manner ?

We thank the reviewer and have reflected on the possibility to do so.

However, in the end we did not feel we could do it in a more concise manner. The use of PI3K and FGFR inhibitors as well as CDK4/6 inhibition is fairly new and these treatments have not been used for all cancers so far. Moreover, the side effects vary, and can be dealt with differently also dependent on patient age. So we prefer just to leave the text as it is.

Again, we thank the reviewers for the valuable comments and hope the revised manuscript is adequately modified

Reviewer 2 Report

In this review, Näsman et al. detail the current knowledge of different HPV+ head and neck cancer subtypes and the promising future treatment options for these. This is an informative review and is worth of publication - I just have a few comments that could improve the current manuscript. Furthermore, the manuscript should be careful edited for language and grammar.

  • Throughout the manuscript, more detailed discussions on how HPV can drive changes in signalling pathways/protein expression should be mentioned in relation to the targetting of these pathways for therapeutics. For example, HPV E6/E7 can induce PI3K/AKT signalling (Morgan et al., Plos Pathogens, 2019; Menges et al., Cancer Research, 2006) and AKT/mTOR (Spangle et al., Plos Pathogens, 2010).
  • A major research drive in HNSCC currently is looking at modulation of cell death signalling through SMAC mimetics. Examples such as birinapant and Debio-1143 are in clinical trials for these cancers. The studies involving these compounds would add to this review immensely (Eytan et al., Cancer Research, 2016; Sun et al., Lancet Oncology, 2020; reviewed in Morgan et al., Cancers, 2020). 

Author Response

Dear Reviewer,

Thank you for your valuable comments. Below, we have answered them point by point.

In this review, Näsman et al. detail the current knowledge of different HPV+ head and neck cancer subtypes and the promising future treatment options for these. This is an informative review and is worth of publication - I just have a few comments that could improve the current manuscript. Furthermore, the manuscript should be careful edited for language and grammar.

  • Throughout the manuscript, more detailed discussions on how HPV can drive changes in signalling pathways/protein expression should be mentioned in relation to the targetting of these pathways for therapeutics. For example, HPV E6/E7 can induce PI3K/AKT signalling (Morgan et al., Plos Pathogens, 2019; Menges et al., Cancer Research, 2006) and AKT/mTOR (Spangle et al., Plos Pathogens, 2010).

We thank the reviewer for bringing these interesting papers to our attention.

The papers of Morgan et al Plos Pathogens and Menges et al., Cancer Research, 2006 pinpoint interesting progress in the cervical cancer field that could be applied to HPV+ OPSCC, since in general, the cervical cancer field is one step ahead.

However, it one cannot always extrapolate, that what is relevant for cervical cancer will apply to HPV+ OPSCC and as also shown in our review that this is not always the case, please see section 6.2. reference [140]. So despite these papers are interesting we have not added them to this review.

Also the paper of Spangle et al., Plos Pathogens, 2010, is very interesting. However, here the authors use the 293T cell line which includes SV40 LT and the UOS20 cell line which is a bone osteosarcoma, so the cell systems although interesting from a basic science point of view are somewhat artificial for clinical extrapolation.

The language of the manuscript has been edited, by a colleague who is English speaking (US) citizen.

  • A major research drive in HNSCC currently is looking at modulation of cell death signalling through SMAC mimetics. Examples such as birinapant and Debio-1143 are in clinical trials for these cancers. The studies involving these compounds would add to this review immensely (Eytan et al., Cancer Research, 2016; Sun et al., Lancet Oncology, 2020; reviewed in Morgan et al., Cancers, 2020). 

We thank reviewer for this information and quoted the review of Morgan et al Cancers et al 2020, at the end of the introduction of section 6.

Also, Eytan et al Cancer Research 2016 was of interest, but only one HPV+ cell line was examined in detail and this cell line UM-SCC-47, showed modest sensitivity to birinapant and death ligands, so it was not cited as such here. Likewise, the trial of Sun et al Lancet Onclogy 2020 was also interesting, but unfortunately, the HPV+ OPSCC group is not described in detail, so one cannot conclude what specific effects Debio-1143 had in this group, so neither was this paper quoted separately.

We hope our replies have been adequate and thank you again for your review.